

# Tricks of the puppet masters: morphological adaptations to the interaction with nervous system underlying host manipulation by rhizocephalan barnacle *Polyascus polygeneus*

Anastasia Lianguzova[1,2], Natalia Arbuzova[1,2], Ekaterina Laskova[1], Elizaveta Gafarova[1], Egor Repkin[1,3], Dzmitry Matach[1], Irina Enshina[1] and Aleksei Miroliubov[2]

[1] Department of Invertebrate Zoology, St. Petersburg State University, St Petersburg, Russian Federation
[2] Laboratory of Parasitic Worms and Protists, Zoological Institute of the Russian Academy of Science, St Petersburg, Russian Federation
[3] Research Park, Center for Molecular and Cell Technologies, St. Petersburg State University, St Petersburg, Russian Federation

Corresponding author
Anastasia Lianguzova, anastasialianguzova@gmail.com

## ABSTRACT

**Background.** Rhizocephalan interaction with their decapod hosts is a superb example of host manipulation. These parasites are able to alter the host's physiology and behavior. Host-parasite interaction is performed, presumably, *via* special modified rootlets invading the ventral ganglions.

**Methods.** In this study, we focus on the morphology and ultrastructure of these special rootlets in *Polyascus polygeneus* (Lützen & Takahashi, 1997), family Polyascidae, invading the neuropil of the host's nervous tissue. The ventral ganglionic mass of the infected crabs were fixed, and the observed sites of the host-parasite interplay were studied using transmission electron microscopy, immunolabeling and confocal microscopy.

**Results.** The goblet-shaped organs present in the basal families of parasitic barnacles were presumably lost in a common ancestor of Polyascidae and crown "Akentrogonida", but the observed invasive rootlets appear to perform similar functions, including the synthesis of various substances which are transferred to the host's nervous tissue. Invasive rootlets significantly differ from trophic ones in cell layer composition and cuticle thickness. Numerous multilamellar bodies are present in the rootlets indicating the intrinsic cell rearrangement. The invasive rootlets of *P. polygeneus* are enlaced by the thin projections of glial cells. Thus, glial cells can be both the first hosts' respondents to the nervous tissue damage and the mediator of the rhizocephalan interaction with the nervous cells. One of the potential molecules engaged in the relationships of *P. polygeneus* and its host is serotonin, a neurotransmitter which is found exclusively in the invasive rootlets but not in trophic ones. Serotonin participates in different biological pathways in metazoans including the regulation of aggression in crustaceans, which is reduced in infected crabs. We conclude that rootlets associated with the host's

nervous tissue are crucial for the regulation of host-parasite interplay and for evolution of the Rhizocephala.

## INTRODUCTION

Host manipulation is a phenomenon widespread across diverse parasitic taxa (*Helluy & Holmes, 2005*; *Poulin, 2010*; *Lafferty & Shaw, 2013*; *Hughes & Libersat, 2019*). Parasites with complex life cycles (such as apicomplexans, digeneans, and acanthocephalans) often impact the behavior of the intermediate host to increase the probability of encountering the next host (*Shaw et al., 2009*; *Cézilly et al., 2010*; *Poulin, Hammond-Tooke & Nakagawa, 2012*; *Hammoudi & Soldati-Favre, 2017*; *Fayard et al., 2020*). In insect parasitoids, host manipulation aimed at protecting the parasite pupae from danger is frequently called bodyguard manipulation (*Lefèvre et al., 2006*; *Libersat, Delago & Gal, 2009*; *Poulin, 2010*; *Dheilly et al., 2015*). Parasitic barnacles are also known for their outstanding abilities to manipulate hosts. For example, these parasites are able to cause feminization of infested male hosts. Namely, infected males become more similar morphologically to females and exhibit behavior patterns of ovigerous females (*Ritchie & Høeg, 1981*; *Hoggarth, 1990*; *Innocenti, Vannini & Galil, 1998*; *Yamaguchi, Aratake & Takahashi, 1999*; *Kristensen et al., 2012*; *Corral, Henmi & Itani, 2021*). Thus, infected crabs take care of the parasite externa (reproductive body containing embryos and larvae) as of their own brood (*Rasmussen, 1959*; *Bishop & Cannon, 1979*; *Takahashi, Iwashige & Matsuura, 1997*). In addition, infected crabs of both sexes were reported to be less aggressive, reducing the chance of externa's damage (*Innocenti, Pinter & Galil, 2003*; *Vázquez-López et al., 2019*). The exact mechanisms of such singular host-parasite interactions remain enigmatic. Nevertheless, specific sites of direct contact with the host's nervous system have been identified (*Nielsen, 1970*; *Payen et al., 1981*; *Miroliubov et al., 2020*; *Lianguzova et al., 2021*).

In the host's hemocoel there is an extensive net of trophic rootlets (the interna) (*Reinhard, 1942*; *Hubert, Payen & Chassard-Bouchaud, 1979*; *Bresciani & Høeg, 2001*; *Noever, Keiler & Glenner, 2016*). These rootlets rarely penetrate and mechanically damage any host's organ systems (*Høeg, 1995*), but the notable exception is the host's nervous tissue. The invasion by interna's rootlets of ventral ganglia is apparently paramount for, at least, the rhizocephalan-decapod system. In basal "kentrogonid" families of parasitic barnacles, the distal part of the invasive rootlets is modified into goblet-shaped organs located in the ganglion periphery (*Nielsen, 1970*; *Payen et al., 1981*; *Miroliubov et al., 2020*; *Lianguzova et al., 2021*). In the only studied "akentrogonid" rhizocephalan, *Diplothylacus sinensis* (Keppen, 1877) (family Thompsoniidae), the goblet-shaped organs are absent; instead, there are numerous heterogeneous rootlets in the ganglion neuropil (*Miroliubov et al., 2023*). "Akentrogonids" are believed to have a lesser impact on their hosts (*Ritchie & Høeg, 1981*; *Høeg, 1995*).

Therefore, goblet-shaped organs characterize the basal "kentrogonids" but are lost in some of the crown "akentrogonids". However, there is no information concerning representatives of the family Polyascidae—the "kentrogonid" rhizocephalans that are a sister group to most of the "akentrogonids" (*Høeg et al., 2020*). Representatives of this family, *e.g.*, *Polyascus polygeneus*, cause conspicuous changes in the host morphology and behavior (*Takahashi, Iwashige & Matsuura, 1997*; *Yamaguchi & Aratake, 1997*).

In this study, we provide a morphological examination of *P. polygeneus* specialized invasive rootlets penetrating the host's nervous tissue, which can elucidate the details of the host-parasite interaction and the evolution of these bizarre parasites.

## MATERIALS & METHODS

### Specimen collection

Asian shore crabs *Hemigrapsus sanguineus* (De Haan, 1835) (in total, 44 specimens) infected with the rhizocephalan *P. polygeneus* were collected in the Sea of Japan (Marine Biological Station "Vostok" of the National Scientific Center of Marine Biology, N: 42.893720, E: 132.732755). A total of 18 specimens for histology and confocal scanning microscopy were collected during July–September in 2017–2020. Some specimens were deposited in the collection of parasitic crustaceans in the Zoological Institute RAS. In August–September 2020–2021 26 infected crabs were sampled for transmission electron microscopy. The parasitised hosts were dissected in sea water under a stereomicroscope MBS-10 (LOMO, Petrograd, Russia). The ventral ganglionic mass with the parasite rootlets were isolated.

### Histology and light microscopy

The host's nervous tissue with *P. polygeneus* rootlets was fixed in Bouin solution (70% saturated solution of picric acid in ethanol, formalin 24% and acetic acid 4.5%) or in Zenker fixative (mercuric chloride 5%, potassium dichromate 2.5%, sodium sulfate 1%, and glacial acetic acid 5% in distilled water). After 2 h fixation and 2 h washing in water, the specimens fixed in Zenker solution were incubated in 70% ethanol with iodine for 1 h, then transferred into 70% ethanol. Then all samples were dehydrated in graded ethanol series and embedded in Histomix™ medium (BioVitrum, St. Petersburg, Russia). Section 5 μm thick were cut using a Leica RM-2265 microtome in the Resource Centre "Molecular and Cell Technologies" (Research Park of St Petersburg University). The sections were stained with Ehrlich's hematoxylin and eosin or with Mallory's trichrome stain, then dehydrated and mounted in BioMount medium (Bio Optica, Milan, Italy). The sections were examined using a Leica DM2500 microscope (Leica Microsystems, Jena, Germany) equipped with a Nikon DS Fi3 camera (Nikon, Tokyo, Japan).

### Transmission electron microscopy

The ventral ganglionic mass and the parasite rootlets were fixed in 2.5% glutaraldehyde in phosphate buffer saline with addition of sodium chloride (pH 7.4, 870 mOsm) at 4 °C. Then the specimens were postfixed in 1% $OsO_4$ in the same buffer (870 mOsm, 1 h). After that the samples were washed in the phosphate buffer saline and distilled water, dehydrated in gradual ethanol and acetone series, and embedded in Epon-812-Araldite embedding

medium. Ultra-thin (63 nm) sections were cut with the use of the diamond knife Diatome "Histo Jumbo" and a Leica EM UC6 ultratome in the "Taxon" Research Resource Center (https://www.zin.ru/ckp/) of the Zoological Institute of the Russian Academy of Sciences, stained with uranyl acetate followed by Reynolds lead nitrate, and examined under a transmission electron microscope Jeol JEM 1400 in the Resource Center "Molecular and Cell Technologies" (Research Park of St Petersburg University). The cuticle measurements were conducted using Fiji ImageJ software (*Schneider, Rasband & Eliceiri, 2012*).

## Confocal laser scanning microscopy

For immunohistochemical visualization the host's nervous tissue and rhizocephalan rootlets were fixed with 4% paraformaldehyde (PFA; Sigma-Aldrich, St. Louis, MO, USA) in phosphate-buffered saline (PBS; Fluka) at 4 °C for 4–8 h. Before the staining, the specimens were rinsed with PBT several times in the course of 24 h (PBS + 0.1% Triton-X100; Sigma-Aldrich, St. Louis, MO, USA). Then the material was incubated with the following primary antibodies solutions: rabbit anti-serotonin (Cat #S5545; Sigma Alrich, St. Louis, MO, USA) (1:1000) and mixture of mouse anti-tyrosinated $\alpha$-tubulin (Cat #T9028; Sigma Alrich, St. Louis, MO, USA) (1:1000), mouse anti-choline acetyltransferase (#AB144P; Sigma Alrich, St. Louis, MO, USA) (1:500) and rabbit anti-VGLUT2 (Cat # V2514; Sigma Alrich, St. Louis, MO, USA) (1:200) or rabbit anti-GABA (#A2052; Sigma Alrich, St. Louis, MO, USA) (1:500) overnight at 4 °C. The samples were washed four times (for 20 min each time) in PBS, and incubated for 12 h at 4 °C with a 1:1000 dilution of anti-rabbit IgG antibodies labeled with Alexa Fluor 488 (Molecular Probes, Cat #A21202) and anti-mouse IgG CFTM 633 (SAB4600138; Sigma Aldrich, St. Louis, MO, USA). The material was then rinsed three times for 10 min each time in PBS, stained with the DAPI nuclei stain (1 ug/ml; Sigma Aldrich, St. Louis, MO, USA) in PBS for an hour, rinsed with PBS and mounted in DABCO-glycerol (1,4-diazabicyclo-[2,2,2-octane]-glycerol).

Specimens for cryosectioning were rinsed with PBT several times in the course of 24 h. Then the material was incubated for 48 h in 15% sucrose solution and for the same time in 30% sucrose solution for cryoprotection. Sections (30–40 μm) were made with the Leica CM3050S cryotome in the "Taxon" Research Resource Center (https://www.zin.ru/ckp/) of the Zoological Institute of the Russian Academy of Sciences. After that, the above-described protocol was used for antibody staining.

All the specimens were examined using the confocal laser scanning microscope Leica TCS SP5, in the "Microscopy and microanalysis" Research Center of the St Petersburg University. The images were obtained using ImageJ software (Fiji) (*Schneider, Rasband & Eliceiri, 2012*).

## RESULTS

### Morphology of the *P. polygeneus* rootlets

Most of the rootlets of *P. polygeneus* are freely located in the hemocoel of its host, the crab *H. sanguineus* (Fig. 1A), and perform the trophic function. Trophic rootlets consist of two cell layers: the hypodermal and axial layers (Figs. 1A–1C). The hypodermal cells are adjacent to the cuticle that makes up the outer electron-dense and the inner homogeneous layers,

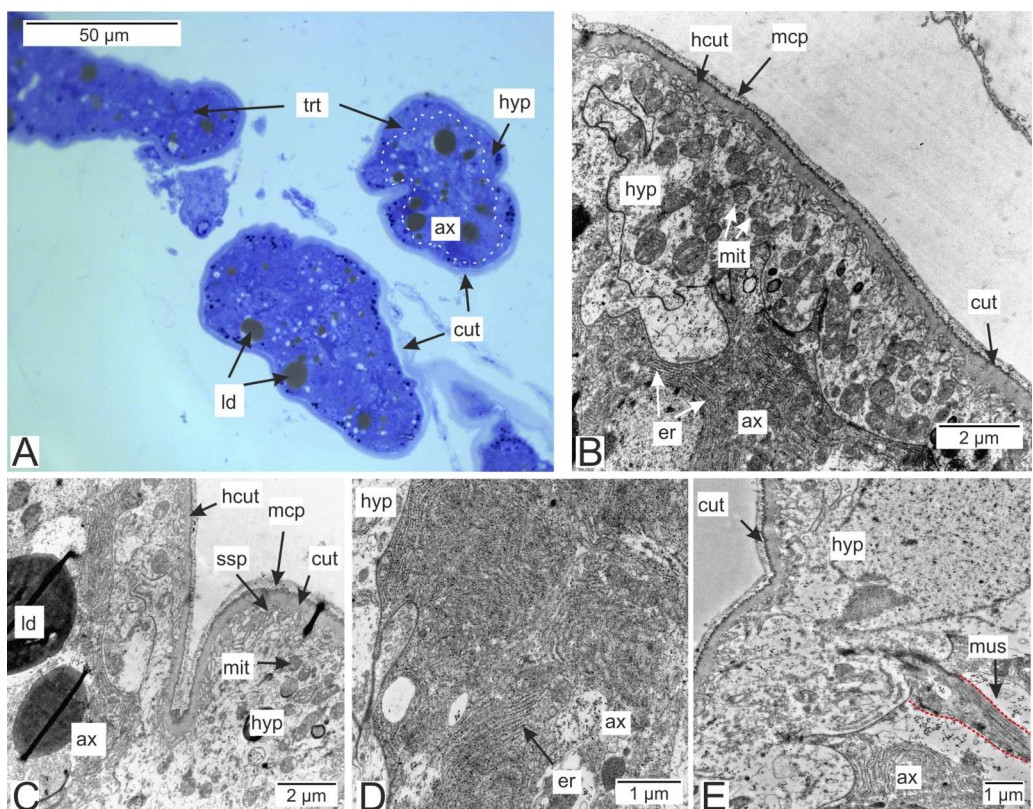

**Figure 1  Interna's rootlets of *Polyascus polygeneus*.** (A) Semithin section of the *P. polygeneus* rootlets. White line marks the border between hypodermal and axial cell layers; (B) ultrathin section of the *P. polygeneus* trophic rootlet with distinct cell layers; (C) *P. polygeneus* trophic rootlet with lipid droplets in the axial cell layer; (D) axial cell; (E) muscle cell in the rootlet. ax, axial cell layer; cut, cuticle; er, endoplasmic reticulum; hcut, homogeneous cuticular layer; hyp, hypodermal cell layer; ld, lipid droplet; mcp, microcuticular projections; mus, muscular cell; ssp, subcuticular space; trt, common trophic rootlet.

and can reach up to 2.8 μm in width (mean 1.36 μm, SD = 0.8 μm). The electron-dense layer forms numerous thin microprojections towards the host's hemocoel (Figs. 1B–1C). The apical cell surface of the hypodermal cells is folded, creating a subcuticular space. The apical zone of hypodermal cells is mitochondria-rich (Figs. 1B and 1C). In comparison with the hypodermal cells, the axial cells have a more electron-dense cytoplasm and contain abundant endoplasmic reticulum and numerous lipid droplets (Figs. 1B–1D). Neither of the rootlet' cell layers possess an underlying basal lamina (Fig. 1D). The center of the rootlets has a prominent central lumen or a lacunar system (Fig. 1E), with muscle cells lining the central lumen border or crossing the central lumen diagonally (Fig. 1E).

In the brachyuran crabs ventral nerve cord is presented by fused ventral ganglionic mass located in the crab's thorax and consisting of both thoracic and abdominal ganglia (Fig. 2A). In infected *H. sanguineus* the parasite's rootlets enlace this part of the nervous tissue (Fig. 2B). Some of the rootlets invade the posterior part of the ventral ganglionic mass (Figs. 2C–2F). These invasive rootlets mostly possess a thin cuticle, one cell layer and a wide central lumen (Figs. 2D–2G). The density of these rootlets can reach a great amount

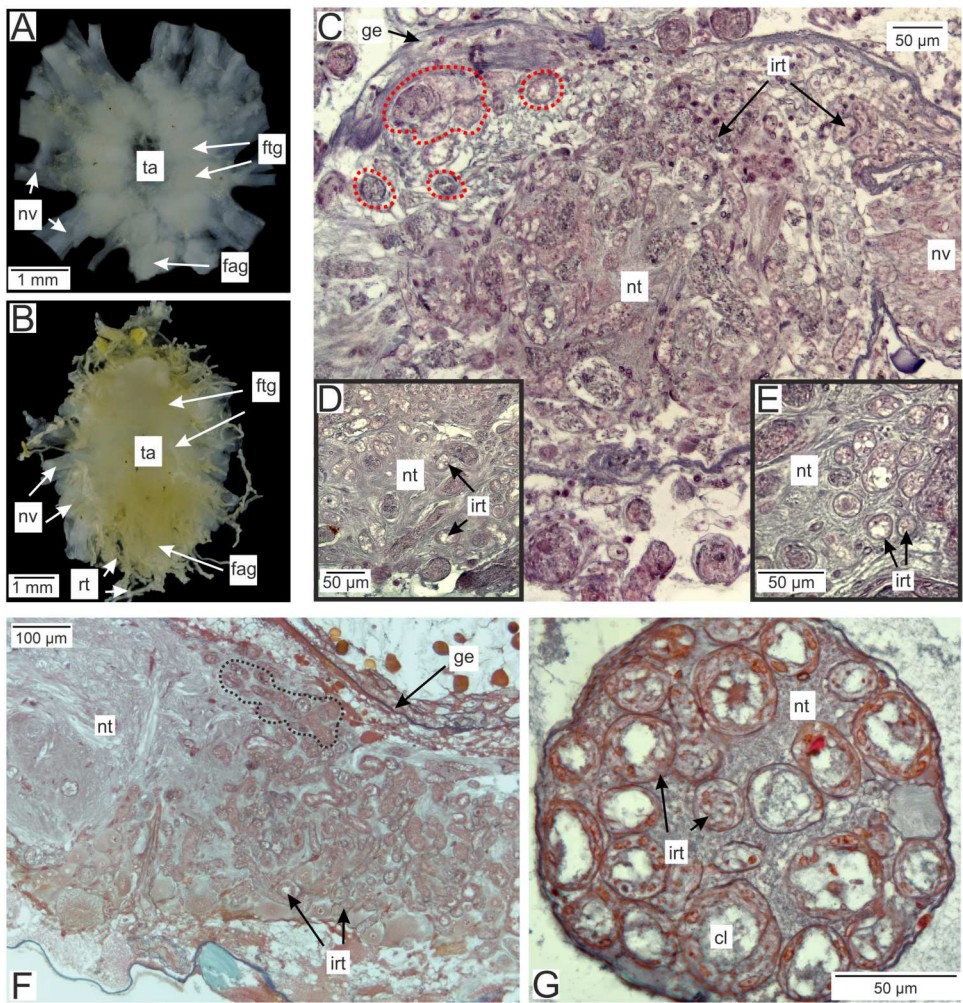

**Figure 2** **Invasive rootlets of *P. polygeneus* in the nervous tissue of the host *Hemigrapsus sanguineus*.**
(A) Dissected ventral ganglionic mass of the healthy crab *H. sanguineus*; (B) dissected ventral ganglionic mass of the *H. sanguineus* parasitized by *P. polygeneus*; (C) cross section of the posterior part of the host's ventral ganglionic mass with numerous invasive rootlets in the nervous tissue (area around some of them is highlighted with red line); (D) area of the ventral ganglionic mass of infected crab at higher magnification; (E) invasive rootlets of *P. polygeneus* in the neuropil of the ventral ganglionic mass; (F) longitudinal section of the ventral ganglionic mass of parasitized *H. sanguineus*. On the left (anterior part of the ganglion) there are few invasive rootlets, on the right (more posterior part of the ganglion) numerous invasive rootlets are present. Area around some of the invasive rootlets is highlighted with black line; (G) nerve of the *H. sanguineus* filled with invasive rootlets. cl, central lumen; fag, fused abdominal ganglia; ftg, fused thoracic ganglia; ge, ganglion envelope; irt, invasive rootlet; nt, nervous tissue; nv, nerves; rt, rootlet; ta, thoracic artery.

in the neuropil of the posterior part of the ventral ganglionic mass (Fig. 2C) but their number is decreasing anteriorly (Fig. 2F). The invasive rootlets of *P. polygeneus* infiltrate not only the ventral ganglionic mass but also the host's nerves (Fig. 2G).

In general, invasive rootlets are thinner than common trophic ones, and can display varying ultrastructural features described below. It is unclear whether these rootlets can
be classified into specific types, or if their ultrastructural features gradually change from one type to another without clear differentiation. For clarity, we will refer to these rootlets as type 1, type 2, and type 3, but it remains uncertain whether these labels correspond to distinct morphological categories. Most of the observed rootlets correspond to type 2 or 3, and on the Fig. 3 we firstly present the scheme of these types.

*Type 1 rootlets.* Type 1 invasive rootlets are rather similar to common trophic rootlets. Like trophic rootlets, they are composed of two distinct cell layers: the hypodermal and axial layers (Fig. 4A). Their cuticle can be divided into the inner homogeneous layer and the outer electron-dense layer (Fig. 4B), which forms microcuticular projections. The cuticle mean width is 376.2 nm (SD = 152.1 nm). The apical membrane of hypodermal cells is folded, and a narrow subcuticular space is still present between the cuticle and the hypodermal cells (Fig. 4B). The axial cells are characterized by the presence of abundant endoplasmic reticulum and lipid droplets (Fig. 4C). Multilamellar bodies associated with membranous organelles (*e.g.*, endoplasmic reticulum) can be observed in type 1 rootlets (Fig. 4C).

*Type 2 rootlets.* Type 2 invasive rootlets are noticeably modified compared to the trophic ones. The wall of these rootlets consists of one cell layer with a wide central lumen (Figs. 2, 5A and 5B). The cuticle of these rootlets is much thinner than in trophic ones (215.7 ± 69.2 nm) and consists of one homogeneous layer that forms tightly packed microprojections towards the nervous tissue (Fig. 5C). These microcuticular projections comprise approximately half of the total cuticle thickness. There is no noticeable outer electron-dense cuticular layer in these rootlets. The cuticle also forms sparse outgrowths into the hypodermal cell layer (Fig. 5C).

Unlike in type 1 rootlets, the apical surface of the hypodermal cells lacks numerous outgrowths, and the cuticle is tightly adjacent to the apical zone with no wide subcuticular space (Figs. 5B–5D). The apical zone is mitochondria-rich (Fig. 5C). The hypodermal cells are connected to each other by desmosomes and septate junctions (Figs. 5C and 5E). We have also observed flattened vesicles adjacent to the lateral cell membranes (Figs. 5C–5E).

Hypodermal cells in the invasive rootlets of type 2 are heterogeneous with at least two cell types present (Figs. 5 and 6A). First, there are cells with electron-lucent cytoplasms full of coated and uncoated vesicles (Figs. 5B, 5D, 5G and 5H). These vesicles are located either in the apical cell zone (Fig. 5D) or in hypodermal cells' outgrowths (Fig. 5G). The second type of cells is characterized by a more electron-dense cytoplasm (Fig. 6A). In these cells, numerous homogenous vesicles and mitochondria are present.

Electron-dense lipid droplets can be found in hypodermal cells and in the cells underlying them (Figs. 5A, 6B and 6C). These lipid droplets differ from the ones found in trophic rootlets: the droplets are irregularly shaped and vary in size, and cytoplasmic invaginations into them are present (Fig. 6C).

We have also observed singular multilamellar bodies in the hypodermal cells of the invasive rootlets (Figs. 5H and 7). Spherical multilamellar bodies of *P. polygeneus* are composed of tightly packed tubular membranes (Figs. 7A–7D). In some multilamellar

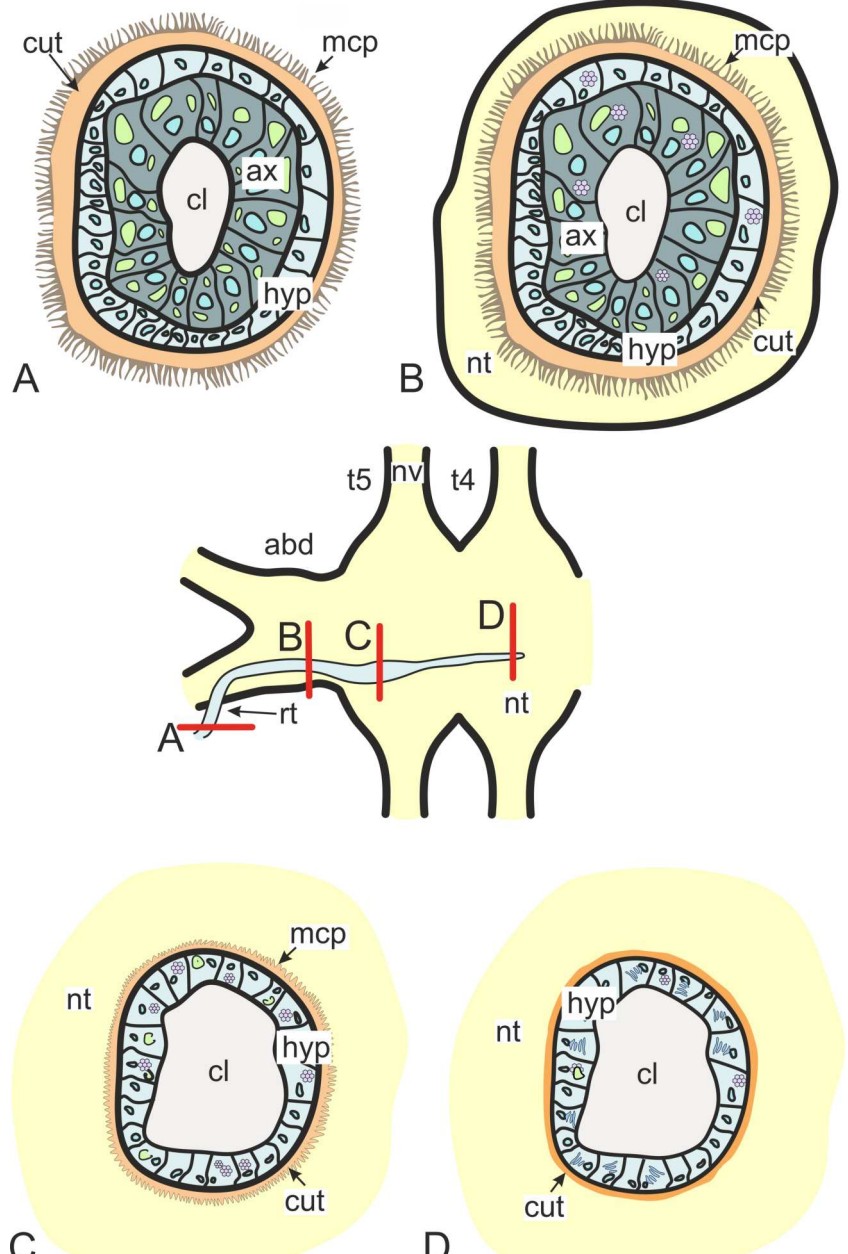

**Figure 3 Transformation of ultrastructural features from trophic rootlets towards the neuropil ones.**
(A) Common trophic rootlet; (B) the first type of the invasive rootlet; (C) the second type of the invasive rootlet; (D) the third type of the invasive rootlets. abd, fused abdominal ganglia; ax, axial cell layer; cl, central lumen; cut, cuticle; hyp, hypodermal cell layer, mcp, microcuticular projections; nt, nervous tissue; nv, nerves; rt, rootlet; t4–t5, fused thoracic ganglia. Green marks lipid droplets, purple marks multilamellar bodies, brown marks electron-dense layer of cuticle, blue marks nuclei.

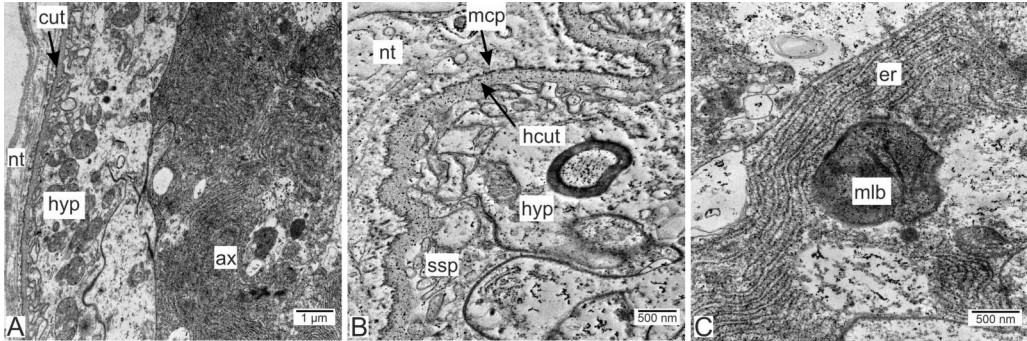

**Figure 4** **Invasive neuropil rootlets of the first type.** (A) General view of the invasive rootlet with hypodermal and axial cell layers; (B) cuticle of the first type of invasive rootlets; (C) multilamellar body in the axial cell. ax, axial cell layer; cut, cuticle; er, endoplasmic reticulum; hcut, homogeneous cuticular layer; hyp, hypodermal cell layer; mlb, multilamellar bodies; mcp, microcuticular projections; nt, nervous tissue; ssp, subcuticular space.

bodies, distinct swollen membranes are present, usually on the lamellar body's border (Fig. 7C). Multilamellar bodies may contact the membrane of lipid droplets or electron-dense bodies (Fig. 7D). In general, these structures are often associated with other membrane organelles in the parasite cells.

*Type 3 rootlets.* Type 3 invasive rootlets are the most modified morphology-wise compared to the trophic rootlets (Figs. 3D and 8). These rootlets possess the thinnest cuticle out of all the discussed types ($58.6 \pm 14.1$ nm), one cell layer, and a prominent central lumen (Figs. 8A and 8B). The cell cytoplasm is electron-dense and filled with tightly packed cisterns of the rough endoplasmic reticulum (Fig. 8B). The apical surface of the hypodermal cells forms finger-like outgrowths parallel to the cuticle which increases the subcuticular space (Fig. 8C). Type 3 rootlet cells contain multilamellar bodies, although they are less common than in type 2 rootlets (Fig. 8B).

Numerous heterogeneous vesicles adjacent to the apical surface of the hypodermal cells and the cuticle can be observed in type 3 rootlets (Figs. 8C–8E). Vesicles cluster near the invaginations of the apical surface, with some clusters appearing in the subcuticular space (Figs. 8C and 8D). The cuticle can generate thin outgrowths, sometimes with bloated distal ends, towards the nervous tissue of the host (Fig. 8E). Tiny electron-dense bodies resembling bloated parasite cuticle outgrowths are present in the nervous cells enlacing the rootlets. However, these bodies are not connected to the parasite cuticle (Figs. 8C and 8D).

*Muscular system of the rootlets.* Muscle cells are present in both the invasive and trophic rootlets of *P. polygeneus* (Fig. 9). Muscle elements are located beneath the hypodermal cell layer (Figs. 9B and 9C). Multidirectional myofibrils, including those perpendicular to each other, can be observed in cells (Fig. 9B). Electron-dense regions connect distinct filaments. Thick filaments surrounded by thin ones are present. Muscle cell outgrowths are conjunct with the hypodermal cells *via* cell contacts. Microtubules connect these cell contacts with

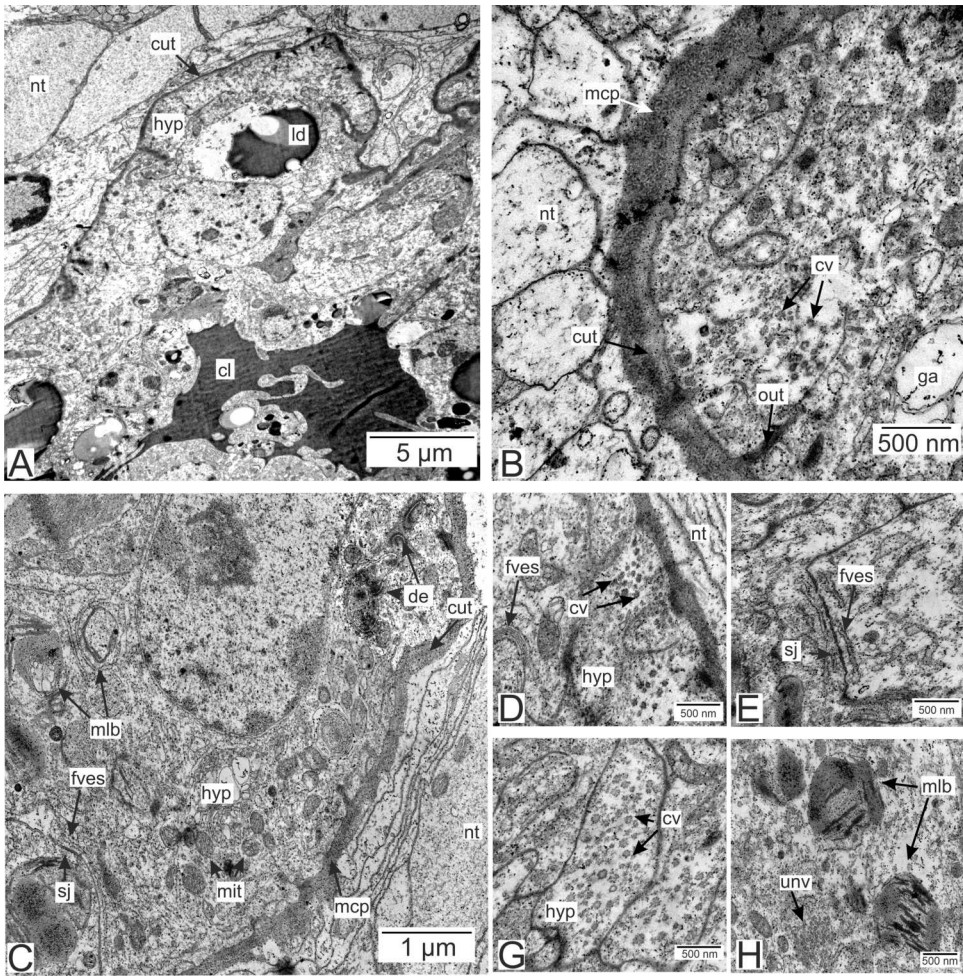

**Figure 5 Invasive rootlets of the second type of *P. polygeneus*.** (A) General view on the invasive rootlet of *P. polygeneus*; (B) cuticle of the invasive rootlet of the second type; (C) general view of the invasive rootlets ultrastructural features; (D) flatten vesicles and septate junction in hypodermal cell layer; (E) coated vesicles in the apical part of the hypodermal cell; (G) coated vesicles in a hypodermal cell outgrowth; (H) uncoated vesicles and multilamellar bodies. cl, central lumen; cut, cuticle; cv, coated vesicles; de, desmosome; fves, flattened vesicles; ga, Golgi apparatus; hyp, hypodermal cell layer; ld, lipid droplet; mcp, microcuticular projections; mit, mitochondria; mlb, multilamellar bodies; nt, nervous tissue; sj, septate junction; out, cuticle outgrowths; unv, uncoated vesicles.

the cuticle outgrowths (Figs. 9C and 9D). A diagram summarizing all the above-mentioned ultrastructural features of the invasive rootlets in Fig. 10.

*The host's nervous tissue.* Processes of nervous cells tightly enlace the parasite invasive rootlets (Fig. 11A). The cuticle in type 3 rootlets forms numerous non-membranous vesicles in the direction of the nervous tissue (Fig. 11B). Similar structures, that are not connected to the cuticle, are associated with these rootlets and can form a layer above the cuticle (Figs. 11A and 11B). Thin host cell outgrowths (presumably of glial cells) enlace most of the rootlets (Figs. 11C and 11D). The cytoplasm of these cells is darker compared

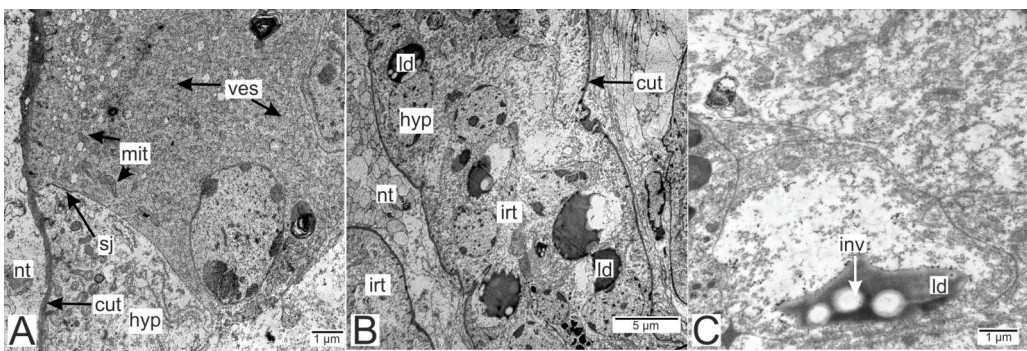

**Figure 6** **Invasive rootlets of the second type of *P. polygeneus*.** (A) Electron-dense hypodermal cell filled with numerous vesicles; (B) lipid droplets spread over the hypodermal cell layer; (C) lipid droplet structure. cut, cuticle; hyp, hypodermal cell layer; inv, invagination of cytoplasm; irt, invasive rootlet; ld, lipid droplet; mit, mitochondria; nt, nervous tissue; sj, septate junction; ves, vesicles.

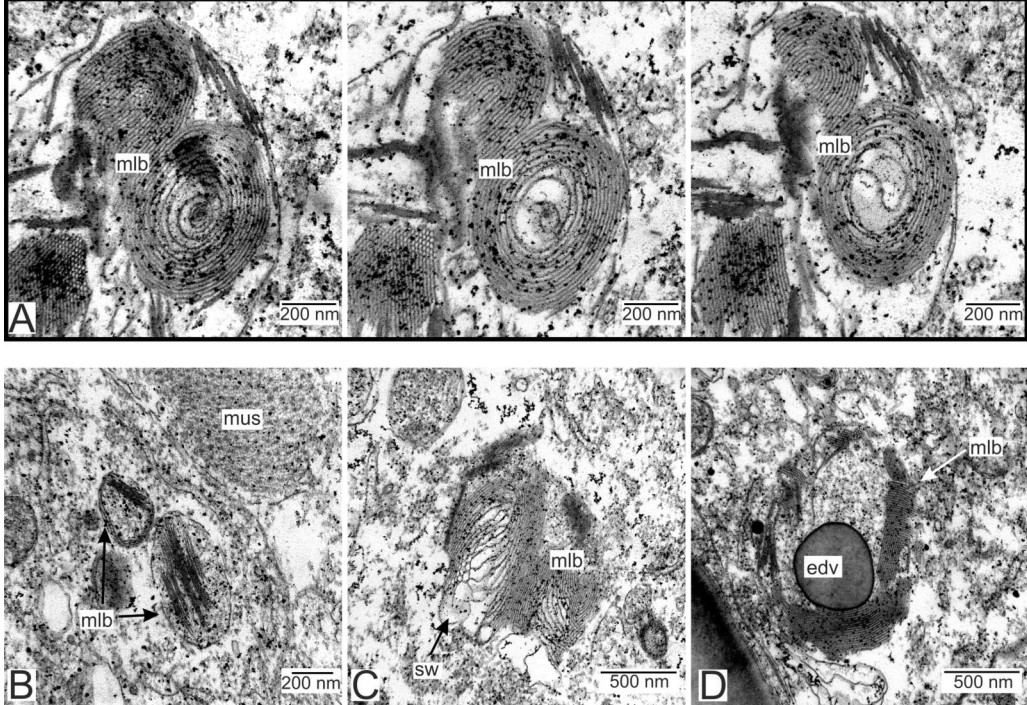

**Figure 7** **Multilamellar bodies in the invasive rootlets.** (A) Membrane packaging in the lamellar bodies in the invasive rootlets of *P. polygeneus* on the series of ultrathin sections; (B) tightly packed membranes in multilamellar bodies; (C) swollen membranes in the lamellar bodies; (D) electron-dense vesicle inside the lamellar body. edv, electron-dense vesicle; mlb. lamellar bodies; mus, muscular cell; sw, swellings of membranes.

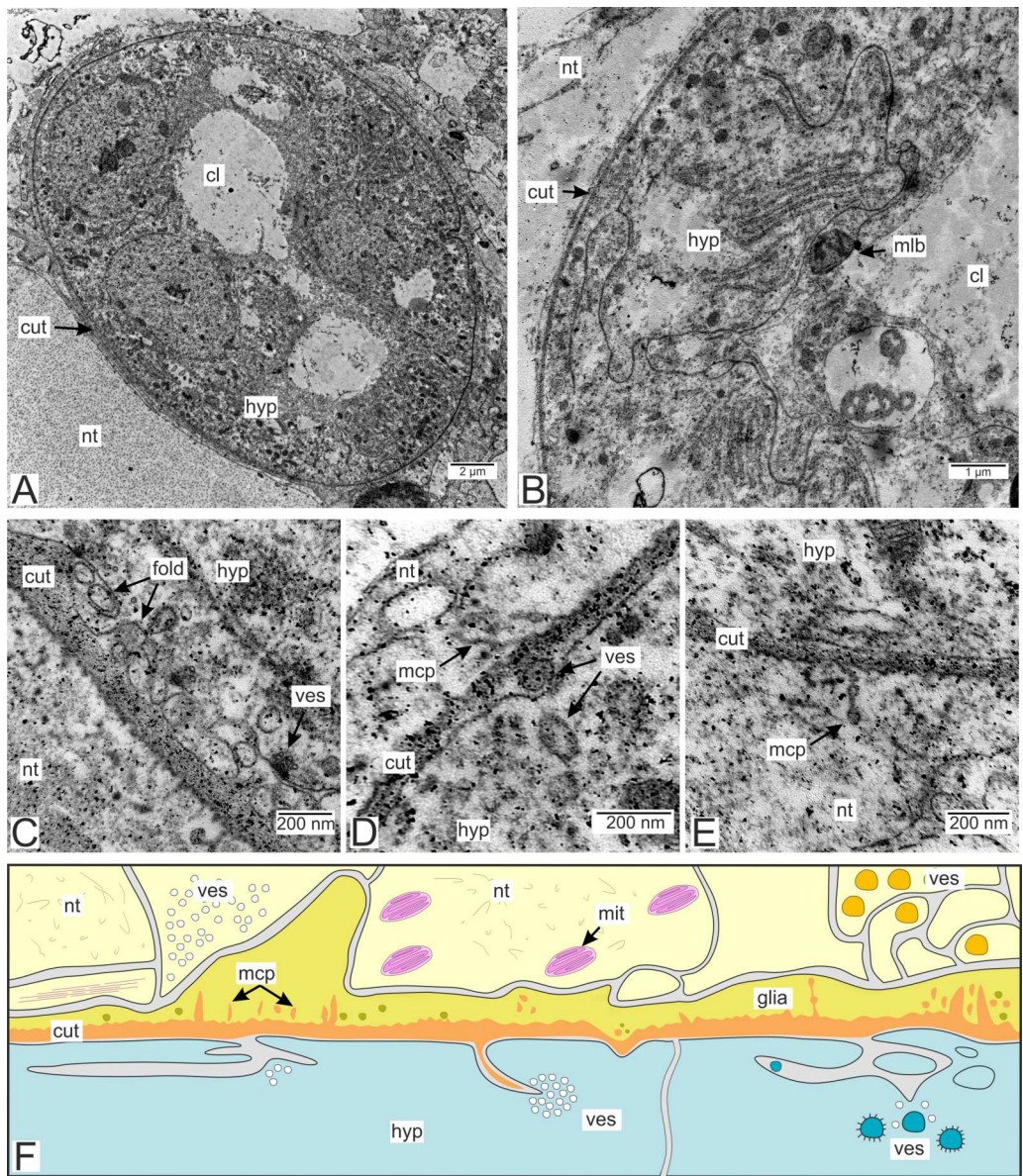

**Figure 8** **The *P. polygeneus* invasive rootlets of the third type.** (A) General view on the invasive rootlet; (B) hypodermal cells of the invasive rootlet of the third type; (C) cuticle of the third type invasive rootlets; (D) vesicles under apical hypodermal cell surface; (E) microcuticular projections towards the host's nervous tissue; (F) scheme of the cuticle and hypodermal cells of the third type. Light yellow marks the host's neurons, vibrant yellow, glial cells adjacent to the parasite cuticle, blue marks the parasite hypodermal cells. Orange is the parasite cuticle. cl, central lumen; cut, cuticle; fold, membrane folds; glia, glial cells; hyp, hypodermal cell layer; mcp, microcuticular projections; mit, mitochondria; mlb, multilamellar bodies; nt, nervous tissue; ves, vesicles.

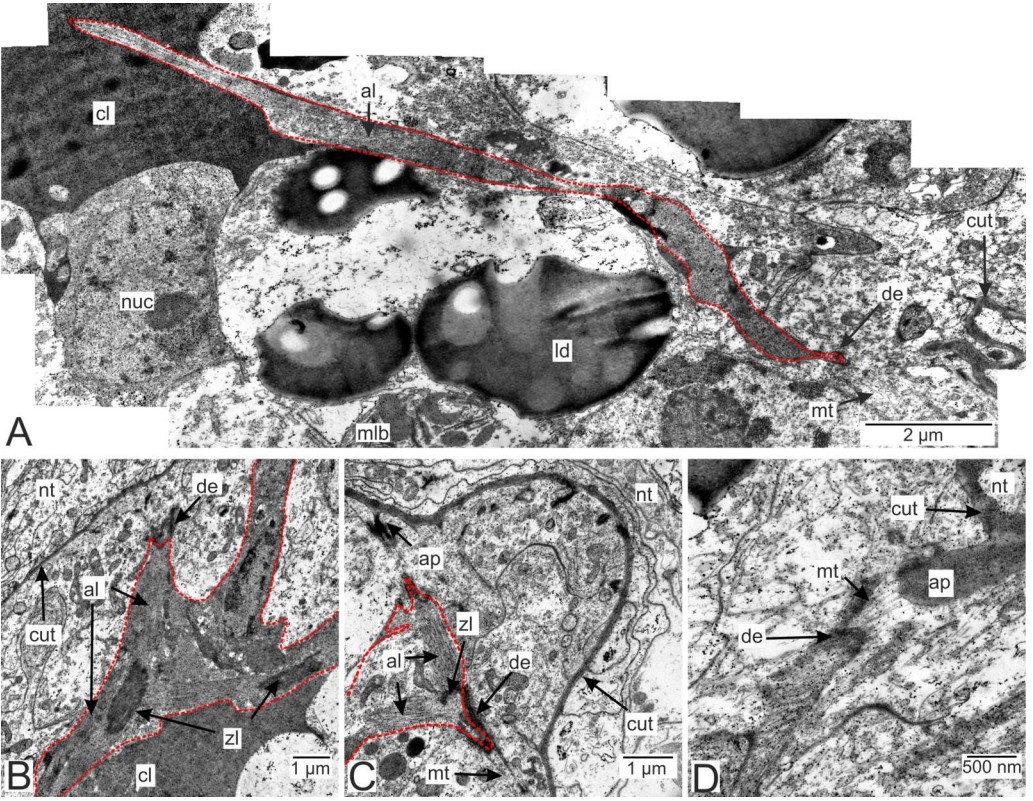

**Figure 9** **Muscular system in the *P. polygeneus* rootlets.** (A) Muscle cell in hypodermal cell layer; (B) muscle cell with myofibrils running in different directions; (C) muscular cell connected to different hypodermal cells; (D) muscular cell connection with the cuticle apodeme *via* microtubules. al, A-line; ap, apodeme; cl, central lumen; cut, cuticle; de, desmosome; ld, lipid droplet; mlb, multilamellar bodies; mt, microtubules; nt, nervous tissue; nuc, nucleus; zl, Z-line.

to the neuronal cells and axons, and the cell membrane is often non-visible, especially adjacent to the parasite cuticle (Figs. 11B, 11D and 11E). Above these cells there are cell processes with electron-lucent cytoplasm filled with microtubules and membranous vesicles (Fig. 11F). In general, the nervous tissue looks intact and functional at the ultrastructural level (Figs. 11, 12A–12C). In most nervous cells mitochondria possess an electron-lucent matrix (Fig. 12C). However, we have also observed modified mitochondria in some cells near the rootlets with swollen cristae and darker matrix (Fig. 12D). Rarely in single cells there were numerous multilamellar bodies (Fig. 12E).

## Antibody staining

We performed antibody staining to reveal some molecules that could potentially be involved in the host-parasite interaction. We compared the invasive rootlets of *P. polygeneus* in the thoracic ganglion and the common trophic ones.

Immunostaining with antibodies to serotonin showed numerous roundish aggregations of it in the invasive rootlets (Figs. 13A and 13B). Serotonin was also found in the nervous fibers enlacing the trophic rootlets, but not in the rootlets themselves (Fig. 13C).

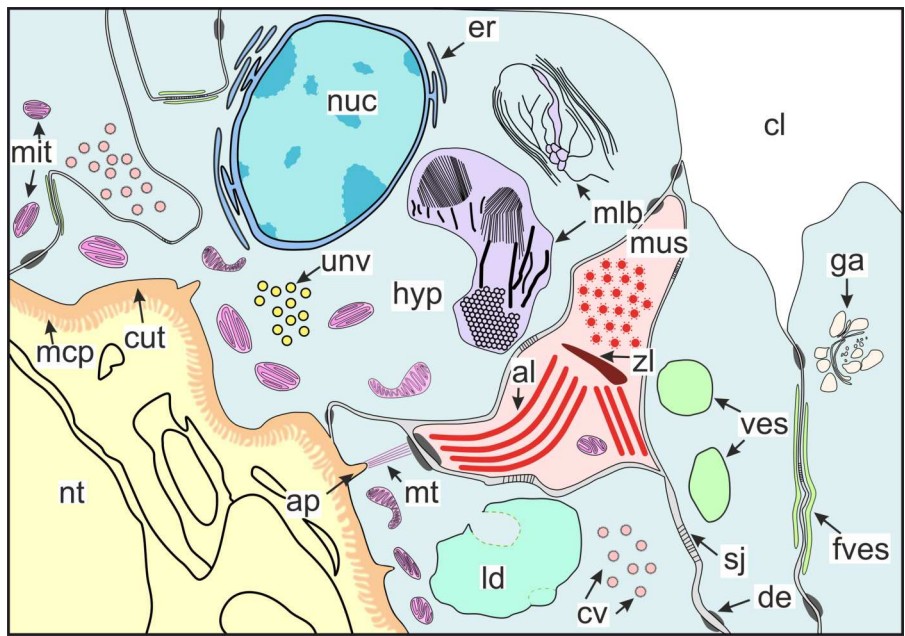

**Figure 10   Scheme of the general organization of the invasive rootlets of the second type.** al, A-line; ap, apodeme; cl, central lumen; cut, cuticle; cv, coated vesicles; de, desmosome; er, endoplasmic reticulum; fves, ûattened vesicles; ga, Golgi apparatus; hyp, hypodermal cell layer; mlb, lamellar bodies; mus, muscular cell; ld, lipid droplet; mcp, microcuticular projections; mt, microtubules; mit, mitochondria; nuc, nucleus; nt, nervous tissue; sj, septate junction; unv, uncoated vesicles; ves, vesicles; zl, Z-line.

Antibodies to Vglut2 (vesicular glutamate transporter) and GABA ($\gamma$-aminobutyric acid) were mapped mostly near the rootlets in the nervous tissue. GABA is mostly present around the rootlets (Fig. 14A) while in the trophic rootlets the signal is absent (Fig. 14B). Cells stained by Vglut2 can envelope like a clutch the rootlets and also can form vast fields near them (Fig. 14C). Trophic rootlets are not noticeably stained with antibodies to Vglut2 (Fig. 14D).

## DISCUSSION

This study utilized transmission electron microscopy and immunocytochemistry to provide a comprehensive account of the interaction site between rhizocephalans and the nervous system of decapod hosts. Using antibody staining, we were able to identify some of the molecules that may contribute to physiological alterations in the host's nervous tissue. Here, we discuss the loss of goblet-shaped organs in *P. polygeneus* and their replacement by neuropil rootlets (rootlets invading the central part of the ganglion) in the evolutionary context of the host-parasite interplay.

### Ultrastructural traits of the *P. polygeneus* neuropil rootlets indicate their primary involvement in the host-parasite interaction

*P. polygeneus* invasive rootlets significantly differ from common trophic rootlets, and possess a number of unique ultrastructural features. The invasive rootlets are also

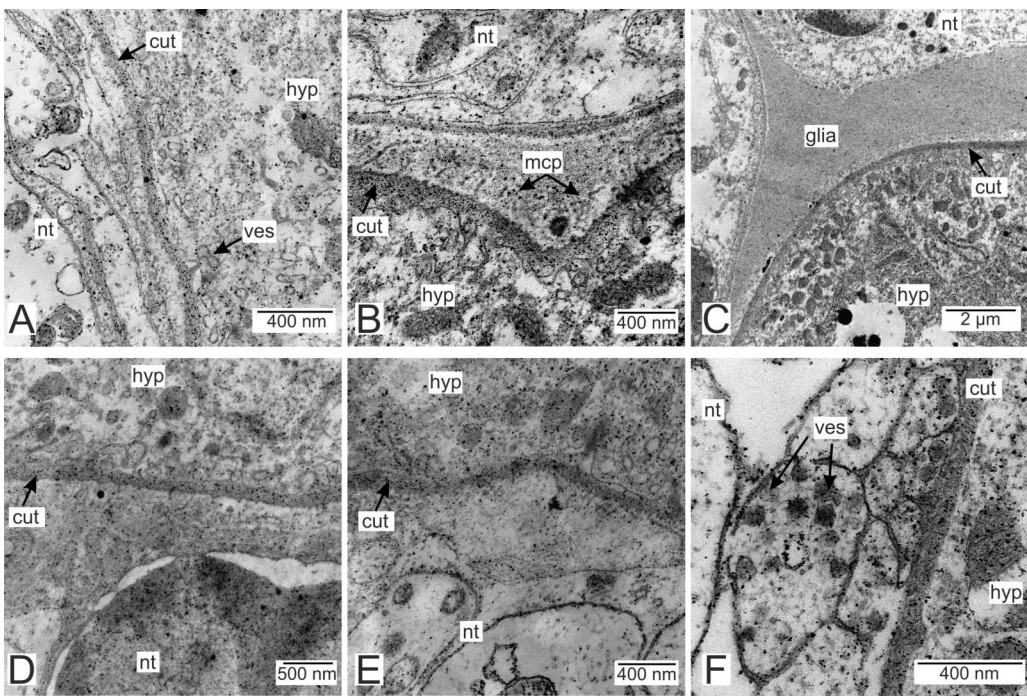

**Figure 11 Nervous tissue adjacent to the P. polygeneus invasive rootlets.** (A) General view on the nervous tissue near the invasive rootlet; (B) numerous microcuticular projections of the *P. polygeneus* towards the host's nervous tissue; (C) general view on the glial cell adjacent to the invasive rootlet; (D) glial cell on high magnification; (E) heterogeneous host's cell adjacent to the parasite rootlet; (F) nervous cells with numerous vesicles adjacent to the invasive rootlet. cut, cuticle; glia, glial cells; hyp, hypodermal cell layer; mcp, microcuticular projections; nt, nervous tissue; ves, vesicles.

heterogeneous in structure. We have identified three distinct types of these rootlets, although it should be noted that this subdivision may not reflect potential gradual morphological changes from one type into another. The first type shares some similarities with trophic rootlets (*e.g.*, cuticle organization and two layers of cells). The second type has only one cell layer, but contains numerous multilamellar bodies, and the cuticle is thinner than in the first type. The third type of the neuropil rootlets is the most modified compared to trophic ones, with the thinnest cuticle and abundant fields of endoplasmic reticulum.

Differences in the ultrastructure of neuropil rootlet types could be explained whether by true presence of the distinct types or by gradual changes along the rootlet's length. Unfortunately, for now we do not have conclusive evidence for any of these hypotheses. However, the presence of transitional forms between these three types more likely implies a gradual change in the ultrastructure along the course of the rootlets.

The multilamellar bodies which we describe here for *P. polygeneus* have previously been reported in the goblet-shaped organs of *Sacculina pilosella* Van Kampen & Boschma, 1925 (Sacculinidae fam.) (*Lianguzova et al., 2021*). Similar structures were formerly observed in the rootlet in the nervous tissue of *Cyphosaccus norvegicus* Boschma, 1962 as well (Peltogasterellidae fam.) (*Bresciani & Høeg, 2001*). Multilamellar bodies of these species differ in morphology, but in all cases form membranous aggregations. It should be

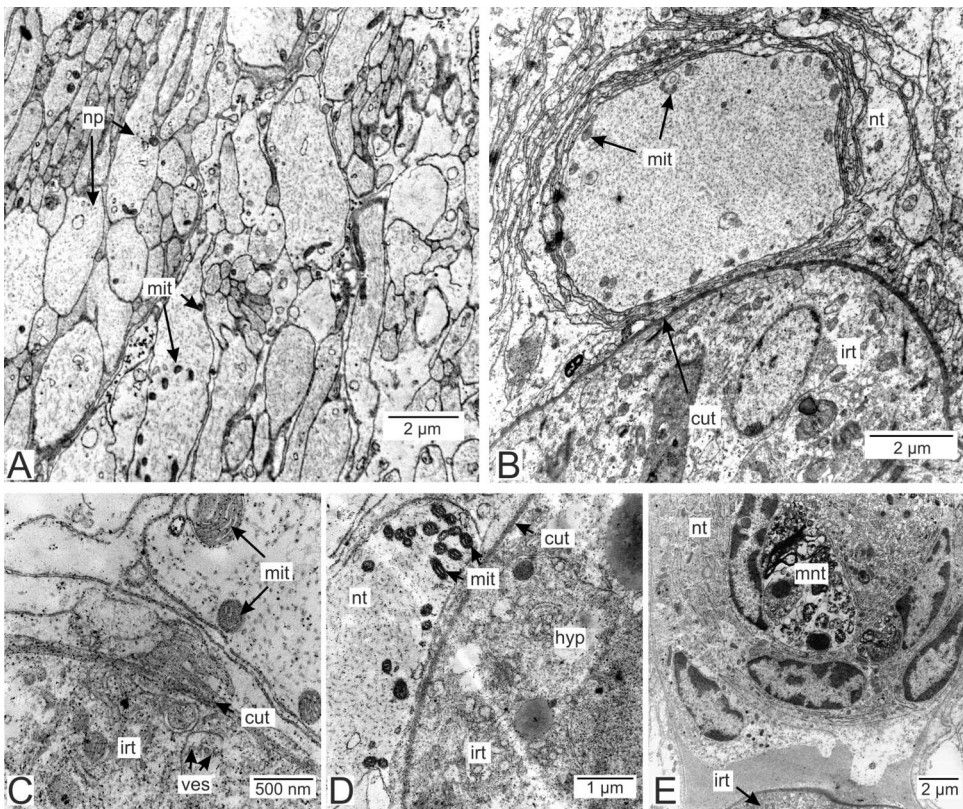

**Figure 12  Nervous tissue due to the *P. polygeneus* invasion.** (A) Non-modified nervous tissue of the infected crab *H. sanguineus* free from the parasite rootlet; (B) non-modified nervous tissue near the invasive rootlets; (C) nervous cell with non-modified mitochondria adjacent to the invasive rootlet; (D) nervous cells with modified mitochondria adjacent to the P. polygeneus; (E) modified nervous cell with multilamellar bodies near the invasive rootlet. cut, cuticle; glia, glial cells; hyp, hypodermal cell layer; irt, invasive rootlet; mit, mitochondria; mnt, modified nervous tissue; np, nervous process; nt, nervous tissue; ves, vesicles.

noted that these structures occur exclusively in the rootlets *P. polygeneus* located in the nervous tissue. *Bresciani & Høeg (2001)* observed "myelin figures" in the *Clistosaccus paguri* Lilljeborg, 1861 ("Akentrogonida": Clistosaccidae) rootlets that resemble multilamellar bodies but are distinct from organelles we found.

Multilamellar bodies are organelles found in a variety of living organisms that are involved in the autophagy processes and can be formed after fusion lipids with lysosomes (*Hariri et al., 2000*; *Lajoie et al., 2005*; *Urbańska & Orzechowski, 2021*). Their presence can be implicated, for instance, in *Caenorhabditis elegans* dauer morphogenesis (Maupas, 1900) Dougherty, 1955 and *Trypanosoma* Gruby, 1843 development through its life cycle (*Meléndez et al., 2003*; *Barquilla & Navarro, 2009*). The multilamellar bodies are also present in the plasmodium of *Intoshia variabili* (Alexandrov & Sljusarev, 1992) and *I. linei* Giard, 1877 (Orthonectida) processes invading the host's nervous tissue (*Slyusarev & Miller, 1998*; *Skalon et al., 2023*). As such, multilamellar bodies in rootlets in the nervous tissue could indicate significant rearrangement of cell machinery and lipid turnover in

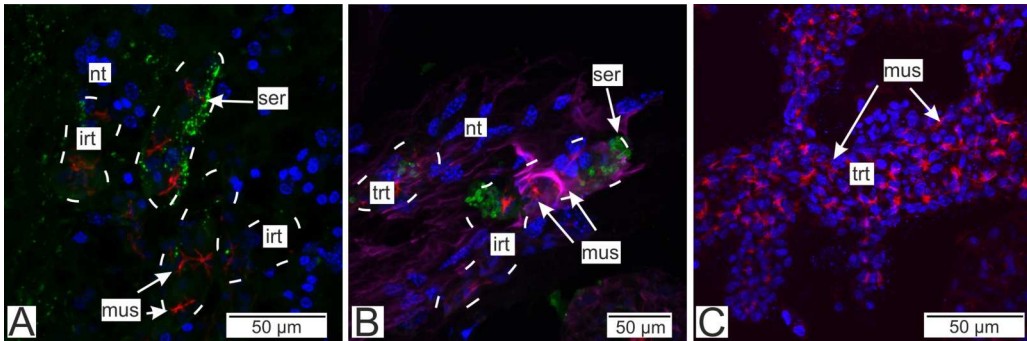

**Figure 13 Distribution of the neurotransmitter serotonin.** (A) Cryosections of the host's ventral ganglionic mass with invasive rootlets of *P. polygeneus* inside (Confocal Z-stack); (B) cryosections of the host's nerve with invasive rootlets of P. polygeneus inside (Confocal Z-stack); (C) whole mount of the common trophic rootlets of P. polygeneus (Confocal Z-stack). irt, invasive rootlet; mus, rootlets' muscular cells; nt, nervous tissue; ser, serotonin aggregations; trt, trophic rootlet. White line marks the invasive rootlet's border. DAPI—blue, phalloidin—red, serotonin—green, alpha-tubulin—magenta.

response to the numerous functions which the invasive rootlets execute. However, the exact functions of the multilamellar bodies in *P. polygeneus* invasive rootlets remain enigmatic.

Lipid droplets occur both in trophic and neuropil rootlets. However, in the neuropil rootlets, the cell cytoplasm can invaginate into the lipid droplets, which has not been observed in trophic rootlets. Lipid droplets are often associated with multilamellar bodies, and can participate in their formation (*Schmitz & Müller, 1991*; *Lajoie et al., 2005*). If so, this could explain the transformation of lipid droplets in rootlets in the nervous tissue, and their consequent replacement by multilamellar bodies.

Morphology of the hypodermal cells is distinct in the neuropil rootlets. We have distinguished at least two distinct cell types: electron-dense cells filled with numerous homogeneous vesicles and mitochondria, and more electron-lucent ones containing coated or uncoated vesicles. These cells probably synthesize different molecules produced by certain cell organelles. Unfortunately, we do not know for certain whether coated and uncoated vesicles are produced by the same cell type or by different ones. Previous studies have identified distinct cells in the hypodermal and axial cell layers of trophic rootlets (*Bresciani & Høeg, 2001*). However, it is unclear whether these cells are truly different types or simply in different stages of the cell cycle. To definitively determine this, a single cell transcriptomic approach is necessary.

In sum, the rootlets penetrating the host's nervous system demonstrate a significant amount of synthetic activity and the potential transport of synthesized substances from the parasite to decapod nervous tissue. The cuticle alterations observed in these rootlets may be attributed to their different function compared to the trophic rootlets. The rootlets located in the host's hemocoel primarily serve trophic purposes, and their cuticle should be permeable to low molecular weight compounds such as glucose, and possibly excreted nerve growth factors (*Miroliubov et al., 2020*; *Nesterenko & Miroliubov, 2022*). On the other hand, the invasive rootlet cuticle likely serves as an intermediary for molecular exchange

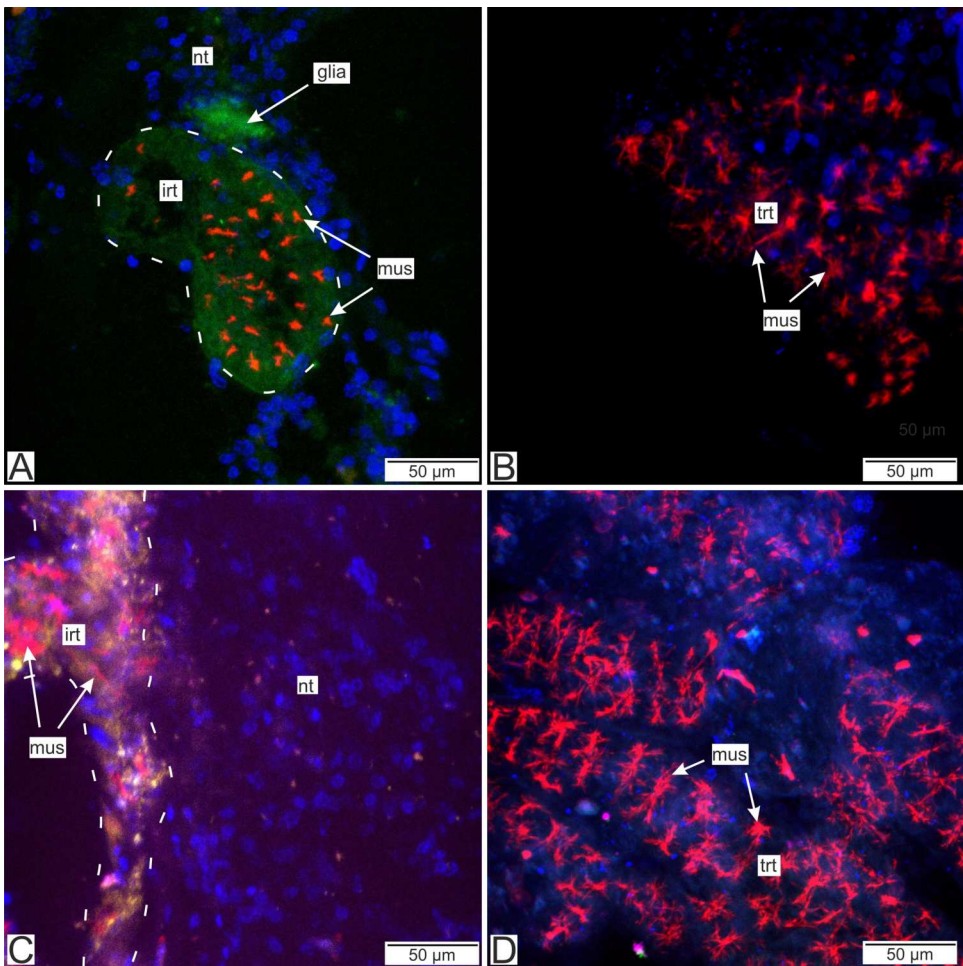

**Figure 14** **Distribution of the glia-associated neurotransmitters.** (A) Cryosection of the host's ventral ganglionic mass with invasive rootlets of *P. polygeneus* inside stained with anti-GABA (Confocal Z-stack); (B) whole mount of the common trophic rootlets of P. polygeneus stained with anti-GABA (Confocal Z-stack); (C) cryosection of the host's ventral ganglionic mass with invasive rootlets of *P. polygeneus* inside stained with anti-VGLUT2 (Confocal Z-stack); (D) whole mount of the common trophic rootlets of *P. polygeneus* stained with anti-VGLUT2 (Confocal Z-stack). glia, glial cell; irt, invasive rootlet; mus, rootlets' muscular cells; nt, nervous tissue; trt, trophic rootlet. White line marks the invasive rootlet's border; DAPI—blue, phalloidin—red, GABA (gamma-aminobutyric acid)—green, VGLUT2 (vesicular glutamate transporter 2)—yellow, CHAT (choline acetyltransferase)—magenta.

between the host's nervous tissue and the parasite, and its organization likely depends on the spectrum of soluble molecules that pass through it.

Moreover, cuticle width in type 3 rootlets can measure approximately 40 nm, and closely resembles the cuticle found on the inner surface of goblet-shaped organs. Such a thin cuticle could facilitate molecular interaction between the host and the parasite. Ultrastructure of the type 3 rootlets also sheds light on some possible pathways of this interaction. In these rootlets, the apical cell membrane forms invaginations and outgrowths, increasing the subcuticular space. Numerous vesicles are associated with these sites; vesicles are also

found between the cell membrane and the cuticle. The cuticle forms projections towards the nervous cells, some of which are shaped as non-membranous vesicles. Such vesicles can also be found disconnected from the parasite's cuticle. Thus, vesicles could be transferred from the hypodermal cells into the subcuticular space, and their content could be released there. The material released from the vesicles is then transmitted to the host's cells in the form of non-membranous vesicles together with cuticle compounds.

*Bresciani & Høeg (2001)* observed vesicles in the apical surface of hypodermal cells and in the homogenous cuticular layer in rhizocephalan trophic rootlets. In the homogenous layer, the vesicles are similar to the invagination of the electron-dense layer of cuticle. The authors did not postulate that these vesicles participate in cross-cuticular transport. Moreover, they indicate that the vesicles are static and embedded in the cuticle during its formation.

Such ultrastructure features as numerous mitochondria, multilamellar bodies and, most importantly, vesicles secreted to the nervous tissue indicate that the neuropil rootlets of *P. polygeneus* located in the nervous tissue of the host could be specifically adapted to the regulation of host-parasite interplay.

## Organization of the muscular system in *P. polygeneus*

Muscle cells were found both in trophic and neuropil rootlets of *P. polygeneus*. The muscular system of this species was described using confocal microscopy (*Miroliubov et al., 2019*). However, only one muscle cell electron micrograph was published previously. In the current study we provide a full description of the ultrastructural features of *P. polygeneus* muscles. Previously described star-like rosettes are possibly distinct cells connected to each other. Crustaceans in general have only striated muscles, and such were seen in confocal images of the *P. polygeneus* (*Miroliubov et al., 2019*). In our electronograms, the muscle elements are contracted and we can distinguish the A-line (the fibers) and Z-line (the electron-dense region) of the sarcomere.

In *P. polygeneus*, the muscle cells establish connections with the hypodermal cells *via* cell contacts. The hypodermal cells contain microtubules that extend and connect with the cuticle outgrowths, which are commonly referred as tendon cells in arthropods (*Reedy & Beall, 1993a*; *Reedy & Beall, 1993b*; *Yamada, 2019*). These cells play a critical role during the molting process as they provide fast muscle connection to the new cuticle. However, rhizocephalans exhibit a lower microtubule density than other arthropods, potentially because molting has never been observed in rhizocephalan rootlets (*Høeg, 1995*). Additionally, the muscular contractions in parasitic barnacles are not as intensive as those in other arthropods.

According to different authors, the muscular system in interna rootlets can perform multiple functions. *Bresciani & Høeg (2001)* proposed that muscle contraction serves to bring new hemolymph to the rootlets. The muscular system can also participate in transporting the nutrients through the central lumen (*Miroliubov, 2017*; *Miroliubov et al., 2019*; *Arbuzova et al., 2022*). The presence of muscles in the invasive rootlets of sacculinids and polyascids could be attributed to the general organization of their interna: the rootlet

system is extensive, and muscular elements are present in every rootlet (*Miroliubov et al., 2019*; *Lianguzova et al., 2021*).

## Interaction with the host's nervous tissue

Our study utilizing immune labeling and CLSM techniques has indicated the presence of certain neurotransmitters or their transporters in the invasive rootlets, while the trophic rootlets lack these substances. Although we cannot yet make precise predictions regarding the mechanisms of host-parasite interplay based solely on immunocytochemistry data, we suggest that these neurotransmitters may potentially play a role in the interaction.

Immunolabeling with antibodies to Vglut-2 (vesicular glutamate transporter) and GABA stained more intensely nervous tissue adjacent to the rootlets. Glutamate and GABA are considered to be markers of the glial cells and mediate glial-neuronal interaction (*Mazaud et al., 2019*). It should be noted that we also detected a signal within the neuropil rootlets but interpretation of this remains obscure. These observations are partly supported by the ultrastructural findings. Rootlets found in the host's nervous tissue are often enlaced by thin processes of cells with electron-dense filamentous cytoplasm. Their ultrastructural organization is similar to glial-like cells. Glial cells are non-neuronal cells that participate in immune, neuroprotection, and neuroregeneration processes (*Ortega & Olivares-Bañuelos, 2020*).

In rodents parasitized by *Toxoplasma gondii* (Nicolle & Manceaux, 1908), enteric glial cells enlace the parasite cysts (*Trevizan et al., 2019*). Astrocytes (a specific type of glial cells) are also involved in triggering tissue inflammation in the early phase of *Trichobilharzia regenti* Horák, Kolářová & Dvořák, 1998 schistosomula infection (*Macháček et al., 2016*). Both microglia and astrocytes are crucial in the repair of nervous tissue after injury (*Chaves da Silva et al., 2013b*; *Antel et al., 2020*), and their presence nearby the rootlets could reflect the host's response to these rootlets invasion.

The proper immune reaction involves subsequent encapsulation of pathogens in the nervous tissue by hemocytes, but these immune cells are extremely rare in case of *P. polygeneus* infection. However, rhizocephalan rootlets could be encapsulated by the host. Such immune response was observed in experimental infection by *Sacculina carcini* Thompson, 1836 of non-specific brachyuran hosts (*Goddard et al., 2005*). It seems probable that rhizocephalans somehow prevent this type of the host response in the established systems. Glial cells can mediate immune and nervous system function in crustaceans (*Freeman & Doherty, 2006*; *Austin & Moalem-Taylor, 2010*; *Chaves da Silva et al., 2013a*).

Glial cells in vertebrate model systems could have a major role in neurodegenerative processes. Neurodegeneration is a final step in a complex of events in the nervous tissue that first include neuroinflammation and mitochondrial dysfunction (*Cragnolini et al., 2020*). We indeed observed altered mitochondria in nervous cells adjacent to the *P. polygeneus* rootlets and modified nervous tissue similar to modification observed in the nervous tissue associated with the goblet-shaped organs of other rhizocephalans species (*Peltogasterella gracilis* (Boschma, 1927) and *Sacculina pilosella*) (*Miroliubov et al., 2020*; *Lianguzova et al., 2021*). Therefore, despite there being no significant visible deterioration of living conditions in infected individuals observed, it is likely that permanent association of the

rhizocephalan rootlets with the host's neurons could possibly lead to degradation of this tissue in parasitized crustaceans.

Another captivating feature of the *P. polygeneus—H. sanguineus* interaction is the presence of serotonin only in the rootlets located in the nervous tissue. Serotonin is exclusively found in the invasive rootlets, indicating its potential role in regulating the interaction between the host and the parasite. Rhizocephalans may excrete serotonin and alter its level in the decapod nervous tissue, or alternatively, the rootlets themselves possibly utilize serotonin as a result of the distinct metabolic processes between invasive and trophic rootlets. However, more research is imperative in order to verify the involvement of serotonin in this particular interaction.

Serotonin is an essential monoamine that has a vital role in many biological processes (*Cools, Nakamura & Daw, 2011*; *Olivier, 2015*). It is mostly considered as a widespread neurotransmitter or hormone that occurs in almost every metazoan taxa (*Turlejski, 1996*; *Caveney et al., 2006*; *Mendl, Paul & Chittka, 2011*; *Kutchko & Siltberg-Liberles, 2013*; *Moroz, Romanova & Kohn, 2021*). In the nervous system serotonin is proposed to have an impact on animal behavior and cognition (*Ellen & Mercer, 2012*; *Kiser et al., 2012*; *Perry & Baciadonna, 2017*). In different invertebrates phyla serotonin regulates the motor and feeding behavior, modulates aggression and anxiety (*Ranganathan, Cannon & Horvitz, 2000*; *Mesquita, Guilhermino & Guimarães, 2011*; *Mosienko et al., 2012*; *Lee et al., 2017*; *Rillich & Stevenson, 2018*; *Bubak et al., 2020*). In crustaceans, in particular, this neurotransmitter increases fighting behavior in subordinate animals (*Huber et al., 1997*). In addition, injection of serotonin into crayfish induces avoidance behavior similar to anxiety behavior of vertebrates (*Fossat et al., 2014*; *Hamilton et al., 2016*). Furthermore, this type of behavior has the survival value to predators at least in amphipods (*Perrot-Minnot, Banchetry & Cézilly, 2017*). However, due to the multifunctional role of this neuromodulator serotonin system can cause opposite effects on animal behavior depending both on diversity of the receptors and the organization of the serotonergic system in general (*Bacqué-Cazenave et al., 2020*).

Serotonin has previously been mentioned as a potential molecule used by rhizocephalans to influence the host (*Vázquez-López et al., 2019*). Parasitic barnacles impact the crabs' agonistic and feeding behaviors, which are known to be regulated by this neurotransmitter (*Bishop & Cannon, 1979*; *Innocenti, Pinter & Galil, 2003*; *Larsen, Høeg & Mouritsen, 2013*). Moreover, we should note that not only rhizocephalans can induce host manipulation *via* serotonin pathway. Thus, acanthocephalans (the parasites in the hemocoel of their intermediate host—the amphipod crustacean) alter serotonin levels in the host brain (*Tain, Perrot-Minnot & Cézilly, 2007*; *Perrot-Minnot, Sanchez-Thirion & Cézilly, 2014*; *Bakker, Frommen & Thünken, 2017*). Although it is too early to draw definite conclusions and future detailed studies are needed, many facts point to the important role of serotonin in this host-parasite system.

## Host-parasite interface through distinct rhizocephalan families

At the moment, specialized contacts of parasitic barnacles with the host's nervous system have been described or at least mentioned for a few of the many rhizocephalan

species. Most of the described cases belong to basal families ("Kentrogonida"): *Peltogaster paguri* Rathke, 1842, fam. Peltogastridae (*Nielsen, 1970*; *Miroliubov et al., 2020*); *Sacculina carcini* (*Rubiliani & Payen, 1979*; *Payen et al., 1981*), *S. pilosella* (*Lianguzova et al., 2021*), fam. Sacculinidae; *Peltogasterella gracilis* (*Miroliubov et al., 2020*), *Cyphosaccus norvegicus* (*Bresciani & Høeg, 2001*), fam. Peltogasterellidae. One case belongs to the crown group ("Akentrogonida"): *Diplothylacus sinensis* (Thompsoniidae) (*Miroliubov et al., 2023*). In basal families, the distal tips of rootlets invading nervous tissue were modified into special goblet-shaped organs. Instead, *D. sinensis* lacks any such morphologically distinct organs.

In the present study, we examined invasive rootlets of *P. polygeneus* from the fam. Polyascidae. All mentioned families, except Thompsoniidae, exhibit a kentrogonid life cycle. However, polyascids are the sister group to crown "akentrogonid" families that are believed to have a lesser impact on their hosts. Our study demonstrates the absence of the goblet-shaped organs in *P. polygeneus* observed in other rhizocephalans. Another polyascid species, *P. planus* (Boschma, 1933), apparently also lacks the goblet-shaped organs (*Hsiao et al., 2016*). The authors have not described the morphology of the rootlets in the nervous tissue; however, neuropil rootlets are visible in the provided photographs.

The neuropil rootlets of the species under consideration, *P. polygeneus*, are heterogeneous in their ultrastructural features. The same applies to the invasive rootlets of "akentrogonid" *D. sinensis* also presented by neuropil rootlets. However, in case of *D. sinensis* this heterogeneity was found even histologically that can possibly point to more significant differentiation of the neuropil rootlets in the "akentrogonids".

Thus, we observed the invasive rootlets in all studied rhizocephalan families. The goblet-shaped organs in the ganglion periphery were presumably lost in the common ancestor of polyascids and the crown "akentrogonids", and replaced by the neuropil rootlets in the central ganglion area. However, this assumption demands further investigations in polyascids and "akentrogonids".

Despite the observed morphological differences, we conjecture that invasive rootlets (both goblet-shaped organs in basal families and neuropil rootlets in crown ones) have similar functions. These structures presumably secrete some soluble molecules directly into the host's nervous tissue.

## CONCLUSIONS

Rhizocephalan parasites induce significant changes in their hosts on physiological and behavioral levels, likely facilitated by invasive rootlets found in the host's nervous tissue. These rootlets, including goblet-shaped organs and neuropil rootlets, are believed to play a crucial role in host manipulation. The absence of goblet-shaped organs in the crown rhizocephalans suggests that they have been replaced by neuropil rootlets in the common ancestor of Polyascidae fam. and "akentrogonids". Invasive rootlets differ from the trophic ones in several key features, including a thin cuticle width, multilamellar bodies, rearrangement of the cell layers, etc. Both goblet-shaped organs and neuropil rootlets appear to secrete and absorb some soluble molecules towards/from the nervous tissue without destroying the host's ganglion tissue. One of the substances potentially participating

in this host-parasite system is the neurotransmitter serotonin, found exclusively in the neuropil rootlets of *P. polygeneus* located in the thoracic ganglion. The host's glial cells may be involved in mediating these interactions. Thus, invasive rootlets may act as a highly specialized neural interface that regulates the communication between parasite and host.

## ACKNOWLEDGEMENTS

We would like to thank O.M. Korn, N.E. Lapshin and the staff of Marine Biological Station "Vostok" of the National Scientific Center of Marine Biology for their help with material collection. The technical support was provided by the staff of the Resource Centers for "Molecular and Cell Technologies" and "Microscopy and Microanalysis" (Research Park of St Petersburg University) and "Taxon" Research Resource Center of the Zoological Institute of the Russian Academy of Sciences. We express our sincere gratitude to the editor and the reviewers for their comments and suggestions, which have significantly contributed to enhancing the quality of our manuscript.

### Funding

This study was funded by grant of the Ministry of Science and Higher Education of the Russian Federation (No. 075-15-2021-1069). Anastasia Lianguzova was funded by a stipend from the Gennady Komissarov Foundation for the Support of Young Scientists. The funders had no role in study design, data collection and analysis, decision to publish, or preparation of the manuscript.

### Grant Disclosures

The following grant information was disclosed by the authors:
Ministry of Science and Higher Education of the Russian Federation: 075-15-2021-1069.
Gennady Komissarov Foundation.

### Competing Interests

The authors declare there are no competing interests.

### Author Contributions

- Anastasia Lianguzova conceived and designed the experiments, performed the experiments, analyzed the data, prepared figures and/or tables, authored or reviewed drafts of the article, and approved the final draft.
- Natalia Arbuzova performed the experiments, authored or reviewed drafts of the article, and approved the final draft.
- Ekaterina Laskova performed the experiments, analyzed the data, prepared figures and/or tables, and approved the final draft.
- Elizaveta Gafarova analyzed the data, authored or reviewed drafts of the article, and approved the final draft.
- Egor Repkin analyzed the data, prepared figures and/or tables, and approved the final draft.

- Dzmitry Matach performed the experiments, prepared figures and/or tables, and approved the final draft.
- Irina Enshina performed the experiments, prepared figures and/or tables, and approved the final draft.
- Aleksei Miroliubov conceived and designed the experiments, analyzed the data, prepared figures and/or tables, authored or reviewed drafts of the article, and approved the final draft.

### Supplemental Information

Supplemental information for this article can be found online at http://dx.doi.org/10.7717/peerj.16348#supplemental-information.

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
