# Peer review of "Tricks of the puppet masters: morphological adaptations to the interaction with nervous system underlying host manipulation by rhizocephalan barnacle Polyascus polygeneus"

_PeerJ, doi:10.7717/peerj.16348_

## Round 0.1 · original submission · Major Revisions

Dear Authors,

I have now received the reviews for your manuscript. While all reviewers see merit in the proposed research they have also suggested a large list of revisions. Many of these are grammatical changes and suggestions to improve the clarity for the reader. However, two of the reviewers have queried the interpretation of some of the histology so please address or rebut these clearly in your revision.

Reviewer 1 ·

Basic reporting

The manuscript “Tricks of the puppet masters: morphological adaptations to the interaction with nervous system underlying host manipulation by rhizocephalan barnacle Polyascus polygeneus” reported a number of TEM sections of the ventral ganglion of an P. polygeneus infested host. The most important finding was that the parasite interna penetrated into and embedded within the host ventral ganglion. This finding means a lot to the study of host control mechanism in parasitic barnacle. In addition, the authors also analysed in detail some ultra-structures such as the cuticle layer in different part of the rootlet system. For instance, the authors reported in detail the differences in the thickness of cuticle layer in different types (or regions) of interna rootlets; they also reported the microvilli-like cuticle projection structures at the outermost surface of the interna, which could have major implication in host-parasitic interactions. I find this paper represents a very important long-lacking piece of work on describing interna this mysterious structure of the mysterious parasite and hence should be accepted for publication.
However, multiple times in the main text, the authors seem to be very subjective when interpreting the TEM data. Given such, there are some major flaws that may need to be fixed with more photo data. I am listing my major comments as followings:

Experimental design

Methods are sufficiently detail.

Validity of the findings

Fig. 1 Based on what criteria did the authors identify infiltrated interna in the neurone shown in Fig. 1D and E? are the black circles in Fig. 1E artificial or staining effect? Irt was not labelled in the figure legend.
Fig. 2 By the cartoon drawing, the authors claimed that interna invade host ganglion cells based on the TEM sections of host fused abdominal ganglia. This is a very important finding that is critical to the understand of host control by rhizocephalans. While in Fig. 1D & E, the author showed two histological sections of interna rootlet embedded inside the ventral ganglion, there is no photo evidence in the paper showing the morphological appearance of the fused abdominal ganglia they sectioned. The authors need to present at least a bright field photo to convince the readers that how the infested host fused abdominal ganglia look like. Was it entangled by interna? Is there a bright field photo showing there is indeed penetration of the interna into the host ganglia as proposed by the cartoon? On a more stringent level (optional, take it as a future work suggestion), the authors should show by biomarker staining (in-situ hybridization that tag host and interna cells) that interna rootlets indeed invaded the ganglia. TEM photos demonstrate very very fine cellular details but at the same time lost large scale spatial information, such that it was actually impossible to tell from the TEM telegraph that which part was host tissue which part was interna. As for now, with just some TEM photos and a cartoon drawing, it is not convincing enough.

Additional comments

The interpretation of the immunostaining part might be misleading. The authors performed immunostaining of serotonin, a neurotransmitter that might be involved in the nervous system of both the host and the parasite. The authors claimed that by detecting GABA, Glutamate and serotonin in the interna invading the host ventral ganglion, they could claim that these neurotransmitters may have a role in host nervous system control. I found such interpretation potentially highly misleading, as the immunostaining result can only tell that the interna contain cell carrying these neurotransmitters, which could be relating to internal regulation of the interna and might completely unrelate to host control. Injection experiment or even neurobiology experimental data will be needed to claim what the authors claiming now.

Line 163, there is no evidence showing in this paper that the hypodermal cell layer secreted the cuticle. Please just state that “a cuticle layer lay adjacent to the hypodermal cell layer”
Line 236, how can the authors tell the non-membranous vesicle in the direction of neural tissue from Fig. 9A?

Reviewer 2 ·

Basic reporting

The manuscript “Tricks of the puppet masters: morphological adaptations to the interaction with nervous system underlying host manipulation by rhizocephalan barnacle Polyascus polygeneus” provides an ultrastructural description of rootlets through which a rhizocephalan exchanges substances with its host. In addition, the presence of selected neurotransmitters in rootlets invading the host’s nervous system is demonstrated via immunohistochemistry.
It is difficult for me to assess the language quality of the manuscript, as I am not a native speaker. I do, however, have some suggestions concerning a few technical terms used by the authors, as pointed out in additional comments.
I find the Introduction and Discussion well organized and clear. The Methods are also well organized but inaccurate and in need of revision. The Results are accompanied by a large number of confocal and electron micrographs of varying quality as well as illustrations, however, the organization of the text and images could be improved. Ultrastructural descriptions are organized according to rootlet type, but the text of these descriptions jumps from one type to another and from one figure to another. Figures are not always referenced in the correct order (e. g., Fig. 2 is referenced after Fig. 3) and sometimes don’t exist (e. g., Fig. 3G). The supposed muscular tissue is covered in a separate section regardless of the structure it is associated with.
Images are mostly provided at low magnifications, which is good for an overview, but very few details can be made out (sepatate junctions are labelled but unclear, coated vesicles similarly). Furthermore, it would be helpful if the structures mentioned in the main text were labelled in the images intended to illustrate these structures.

Experimental design

As the aim of the study is to provide a morphological description of invasive rootlets, the methods used are appropriate. The ultrastructural description provides additional information to what has been published by the authors previously, while the confocal images supplement the electron microscopy by illustrating the shape of the supposed muscle cells with unusual morphology as well as the localization of serotonin and several other molecules.The conclusions of the manuscript are supported by the presented results, but certain ultrastructural features can be given some further consideration. Numerous reported ultrastructural details are not clear in the provided images, as pointed out in additional comments. The reported absence of a basal lamina is unusual, as the epidermis is an epithelium and Figures 1 and 3 indicate that a basal lamina might in fact underly the epidermis. Numerous reported extracellular vesicles and cuticular regions not in contact with the rest of the cuticle could be evaginations connected to the cell membrane or the cuticle outside the plane of the thin section. The reported ultrastructural changes in the host’s nervous system cannot be assessed without comparison with a healthy nervous system of parasite-free crabs.

Validity of the findings

The conclusions of the manuscript are supported by the presented results, but certain ultrastructural features can be given some further consideration. Numerous reported ultrastructural details are not clear in the provided images, as pointed out in additional comments. The reported absence of a basal lamina is unusual, as the epidermis is an epithelium and Figures 1 and 3 indicate that a basal lamina might in fact underly the epidermis. Numerous reported extracellular vesicles and cuticular regions not in contact with the rest of the cuticle could be evaginations connected to the cell membrane or the cuticle outside the plane of the thin section. The reported ultrastructural changes in the host’s nervous system cannot be assessed without comparison with a healthy nervous system of parasite-free crabs.

Additional comments

Line 83: P. polygeneus should be first written in full and later abbreviated. The abstract and figures are standalone parts of the manuscript, so this applies to them separately from the main body.
Line 102: Bouin’s solution does not consist of 70 % picric acid, but a 70 % (by volume) of a saturated solution of picric acid in ethanol; 24% paraformaldehyde is likely not the case, morel likely 24% formalin, which is a 30% solution of formaldehyde, resulting in a much lower formaldehyde concentration.
Line 106: Why was ethanol with iodine used?
Line 108: Sections 5 mm thick were cut using a Leica RM-2265 microtome – 5 mm is an unreasonable thickness for histology sections.
Line 140: What is DABCO-glycerol?
Confocal laser scanning microscopy: Most of the primary antibodies used were rabbit antibodies, whereas the secondary antibodies are reported to be anti-mouse antibodies. This would not allow the detection of primary antibodies.
Line 143: Sections (40 nm) were made with the Leica CM3050S cryotome – 40 nm seems unusually thin for cryostat sections.
Line 147: confocal staining laser microscope – confocal laser scanning microscope
Line 152: Polyascus polygeneus should be in italic.
Line 155: Hemigrapsus sanguineus is always written out but can be abbreviated to H. sanguineus throughout the rest of the manuscript.
Line 162: Hypodermal – epidermis is a more suitable term then hypodermis. While I acknowledge that hypodermis has historically been used to describe the epithelium beneath the cuticle in crustaceans, a more universal term used when referring to the ectoderm-derived integumental epithelium that secretes the cuticle is epidermis.
Line 166: A wide subcuticular space – The membrane is clearly folded, but no space is obvious.
Line 184: Electron translucent – the authors might wish to consider using the term electron lucent.
Line 184: may be filled with electron-dense contents that can surround more electron-translucent spherical spaces – It is clear from figure 4B that a lot of the surrounding cells are lysed and the electron-lucent spherical spaces might simply be cytoplasm leaking from these cells into the central lumen, displacing the electron-dense substance.
Line 189: The cuticle also forms rarely located outgrowths into the hypodermal cell layer – This is not presented on a figure. I suggest sparse, not rarely located.
Line 203: Some of the droplets are lined by a visible membrane (Fig. 3G) – Such an observation requires a clear demonstration at high resolution, as this is unusual for lipid droplets and a misleading appearance of a membrane may in some cases result from contrasting artefacts.
Line 206: Muitilamellar bodies – These multilamellar bodies appear to come both in a lamellar form or as hexagonal lipid-liquid crystals. Their morphology is suggestive of massive lipid turnover, but naturally the functions of such organelles in arthropods are difficult to assess.
Line 210: Lysosomes – how certain are the authors that these are lysosomes?
Line 221: Vesicles cluster near the invaginations of the apical surface, with some clusters appearing in the subcuticular space (Fig. 6C,D) – None of this is apparent in these images.
230: Muscular fibers, including those perpendicular to each other, can be observed in cells – muscle fibers are synonymous with muscle cells, what is likely meant are myofibrils.
Line 241: If the outer membrane is visible, it is often partly disrupted (Fig. 9E) – This is unclear from presented images and requires an explanation. Is it disrupted due to inadequate fixation and processing or cell lysis in vivo?
Line 279: It is possible that interaction with the host’s nervous tissue induces physiological changes in the rootlets – This is highly speculative. What is also not clear from the manuscript is whether there is a change of morphology along the length of a rootlet (as indicated in Figure 2) or there are different types rootlets? This, I think, could be demonstrated by careful examination of sections or confocal microscopy.
Line 347: Electronogram – I suggest electron micrograph.
Line 357: This could be attributed to their active role in the arthropod molting cycle, as microtubules facilitate muscular connection with the cuticle after the molt – This claim lacks a reference and doesn’t make much sense to me. Microtubules in the epidermal cells at points of muscle attachment (“tendon cells”) are always present, mechanically lining the muscle cells to anchors in either the old or the new cuticle. Furthermore, do the invasive rootlets molt at all? And how is this accomplished?
Line 395: Altered mitochondria – Altered in what way and in relation to what?
Line 411: 5-HT – the term 5-HT needs to be explained and either serotonin or 5-HT used consistently throughout the manuscript.
Illustrations: Lines representing cell membranes are too thick in comparison to other lines in the drawings. This is not an objective flaw of the images but makes them less appealing.
Figure captions – Individual images constituting a figure (A, B, C) should also be explained in the caption.
Figure 1: The initial information in text is presented on images D and E, which should therefore come first.
Figure 3A: This figure is referred to after Figure 1 in text, so it should be Figure 2.
Figure 6: Whatever the ga arrow is pointing at, it does not resemble the Golgi apparatus.
Figure 10: the label Glia seems to label intercellular space.

·

Basic reporting

All Basic reporting issues are fine

Experimental design

Experimental design and Materials and Methods are all fine

Validity of the findings

I agree that the Results are based on solid facts

Additional comments

I have been late with this review and thus hurry it along. But all in all I have no major issues at all. The author team is at present the leading authorities on rhizocephalan root systems and use an array of methods with great skill. The are fully aware and cite the relevant literature, including, to my pleasure, the older "classical" works that contain much valuable information.

I could elaborate at length with comments ands recommendations, but none I can offer are really quint essential to the MS. Instead I will give a few points of interest

1) TLine 307-308: The authors mention two epidermal (hypodermal) cell types: Light ones and darker (pigmented) ones. In Clistosaccus paguri (Hoeg, either 1985 or in Bresciani and Hoeg, 2001) also depicted two distinct hypodermal cells types. But these were apparently NOT associated with the host nervous system.

2) I wonder why the authors consistently use the term "hypodermis" rather than "epidermis". I find "hypodermis" somewhat "archaic". But never mind!

3) Line 358: "encapsulation of roots". I seem to remember that the "Goddard" paper mentioned that encapsulation of rhizocephalan roots mostly or exclusively happened when parasites infested decapods that in the field were "non-hosts". ? If true, the lack of encapsulation could indicate an adaptation on part of the parasite to ensure success?

4) Root histology and phylogeny. The authors are well aware that the Akentrogonids as originally defined are not polyphyletic. Yet, they also know that "akentrogonid types" other than Mycetomorpha may still form a monophyletic entity. In this context it is highly interesting if there are structures in thre root system that support such a clade, or a clade of these and the Polyascidae. Clearly, adaptation to the host and host-control is central in rhizocephalan evolution, and it would be surprising if this is not also reflected in root ultrastructure and function. Here we see that this may indeed be the case.

As an aside, THIS REVIEVER may have histological slides of both Mycetomorpha externae AND their internae (roots and internae). Even if only parafin sections, they might reveal if the goblet shaped bodies are present.
The editor can feel free to provide the identity of this reviewer

---

## Round 0.2 · accepted · Accept

Dear Authors,

Thank you for your submission to PeerJ. I'm delighted to inform you that your manuscript has been accepted.